# Methodological Insights in Detecting Subtle Semantic Shifts with Contextualized and Static Language Models

**Sanne Hoeken**[1], **Özge Alaçam**[1,2], **Antske Fokkens**[3,4] and **Pia Sommerauer**[3]

[1]Computational Linguistics, Dept. of Linguistics, Bielefeld University
[2]Center for Information and Language Processing, LMU Munich
[3]Computational Linguistics & Text Mining Lab, Vrije Universiteit Amsterdam
[4]Department of Mathematics and Computer Science, Eindhoven University of Technology
`{sanne.hoeken, oezge.alacam}@uni-bielefeld.de`
`{a.s.fokkens, pia.sommerauer}@vu.nl`

## Abstract

In this paper, we investigate automatic detection of subtle semantic shifts between social communities of different political convictions in Dutch and English. We perform a methodological study comparing methods using static and contextualized language models. We investigate the impact of specializing contextualized models through fine-tuning on target corpora, word sense disambiguation and sentiment. We furthermore propose a new approach using masked token prediction which relies on behavioral information, specifically the most probable substitutions, instead of geometrical comparison of representations. Our results show that methods using static models and our masked token prediction method can detect differences in connotation of politically loaded terms, whereas methods that rely on measuring the distance between contextualized representations are not providing clear signals, even in synthetic scenarios of extreme shifts.[1]

## 1 Introduction

With the increasing popularity and success of language models, Lexical Semantic Change Detection (LSCD) has developed into its own research field. The primary focus has been placed on the automatic detection of extreme semantic shifts across time periods (i.e. *diachronic* shifts). More subtle semantic shifts such as differences in connotations attributed to words by socially different communities have received little attention. Such shifts can, however, provide highly relevant insights for social scientists, as they may reflect fundamental differences in world views. As connotations and associations are likely to be reflected by the contexts of words, static and contextualized language models have the potential to capture them.

The current state of the field does not provide conclusive predictions of the most effective models and methods for detecting subtle shifts in connotation. The best-performing methods for LSCD predominantly rely on static embedding representations. Several methods for contextualized models have been proposed but have generally been outperformed by static methods on LSCD tasks (Kutuzov et al., 2022; Schlechtweg et al., 2020). Incidental results from a Russian and Spanish shared task (Kutuzov and Pivovarova, 2021; Zamora-Reina et al., 2022) show that contextualized models can outperform static models when given the right information through fine-tuning on a task that foregrounds sense information. In this paper, we systematically investigate different methods using contextualized and static models on a use case of subtle semantic shifts in political discourse. We test different approaches to prioritize relevant information in the model and investigate how contextualized models react to information in the input data.

Our experiments yield the following insights: (1) Standard methods relying on contextualized representations fail to detect subtle shifts. We use a data manipulation approach to simulate extreme shifts and demonstrate that none of the methods can detect them. Our experiments confirm the intuition from Schlechtweg et al. (2020) that information learned in pretraining provides such a strong signal that contexts reflecting a new sense of a target token do not impact its representation. (2) The best-performing LSCD system of the Russian and Spanish shared task (Rachinskiy and Arefyev, 2022) did not perform best on our subtle shift task nor our evaluation on the English subtask of SemEval 2020. (3) We propose an alternative method for contextualized models relying on masked token prediction rather than representation comparison inspired by Arefyev and Zhikov (2020). In contrast to the representation-based contextualized methods, this method is able to predict subtle shifts. Our substitution method has the additional advantage that it is more transparent, as the predicted substitutions

---

[1]The code used for the experiments can be found at `https://github.com/SanneHoeken/LSVD`

give insights into the nature of the shifts. In addition to our main insights, we confirm that word form has an impact on the performance of contextualized models (as suggested by Schlechtweg et al. (2020) and examined by Laicher et al. (2021)); if raw test data is provided, instead of lemmatized (which is a de-facto input style for lexical change experiments), BERT outperforms static models on SemEval.

## 2 Background and Related Work

In this section, we first provide an overview of approaches for Lexical Semantic Change Detection (LSCD), the task of automatically detecting shifts in semantics.[2] This is followed by a brief presentation of proposed approaches for validating LSCD methods and a discussion of work that addresses social variation rather than diachronic change.

### 2.1 Semantic shifts

Semantic shifts occur when words undergo a change in meaning. The most well-known scenario for this is diachronic semantic change, where words acquire new meanings over time, while old meanings may disappear. Another type of shift can occur due to social variation; different communities may use words in different senses or ascribe different connotations to them (Newman, 2015). Distributional approaches hold great potential for studying both types of shifts, as shifts in word meaning should be reflected by differences in their contexts within community- or time-specific corpora.

**Static Embeddings**   Count- and predict methods for static embeddings can be used to detect semantic shifts by creating separate models for each corpus. Semantic shifts are detected by either comparing the distance between vector representations of the same word across the target corpora (requiring aligned model spaces) or by comparing the relative positions of a word in the semantic spaces (e.g. by comparing neighborhoods or distances to selected other words) (Hamilton et al., 2016a,b; Kim et al., 2014; Gulordava and Baroni, 2011).

**Contextualized Models**   Pretrained contextualized language models can be used to detect semantic shifts in various ways. One common approach is to use an existing pretrained model to extract

representations of a target word. The representations are then compared across the corpora (either by means of a pairwise comparison or by creating aggregated word type or sense representations) (Giulianelli et al., 2020; Martinc et al., 2020, e.g.). Alternatively, a pretrained language model can also be leveraged for masked token prediction. Arefyev and Zhikov (2020) extract the top k predicted candidates for the target word to induce clusters representing word senses. They then compare the occurrence frequency of these word senses across corpora. In both approaches, the pretrained language model can be fine-tuned to the corpora under investigation by means of continued pretaining (Kutuzov and Giulianelli, 2020; Martinc et al., 2020; Montariol et al., 2021, e.g.), which has been shown to lead to performance gains.

The pretrained (and domain-adapted) model are also further fine-tuned on a specific task to incorporate information relevant to semantic change. So far, a method employing fine-tuning for Word Sense Disambiguation is the only method for contextualized models that has been shown to outperform static models (Rachinskiy and Arefyev, 2022) on two recent shared tasks on LSCD in Russian (Kutuzov and Pivovarova, 2021) and Spanish (Zamora-Reina et al., 2022), respectively. The intuition behind this approach is that the fine-tuning step foregrounds information about word senses rather than word forms. This addresses one of the issues identified by Laicher et al. (2021), which pertains to the orthographic or grammatical bias observed in contextualized language models when performing LSCD.

**Static vs Contextualized**   Methods for both model types rely on the assumption that semantic changes in words are reflected in their contexts. However, they differ fundamentally in terms of how the models are created; while contextualized approaches for LSCD use already pretrained models, static approaches create new models to represent each corpus. Apart from some diachronic approaches where the model for the previous time period is used as a starting point (Kim et al., 2014), they are largely *created* from scratch taking cooccurrences of the full vocabulary into account.

Methods based on contextualized models *extract* instance representations from corpora using pretrained models. The representations thus incorporate information from their pretraining, which remains identical across the corpora being compared.

---

[2]We focus on approaches closest to ours and refer the reader to Kutuzov et al. (2022) and Tahmasebi et al. (2021) or Montanelli and Periti (2023) for more extensive overviews.

Based on our initial experiments, we observed that methods relying on contextualized representations did not reflect subtle shifts at all. Addressing the possible explanation of pretraining dominance, we simulate scenarios of extreme shift via a replacement strategy.

## 2.2 LSCD for Subtle Synchronic Shifts

While LSCD methods have been widely developed for historical purposes, only a few studies examined synchronic semantic shifts. Del Tredici and Fernández (2017) adapt the LSCD method of Dubossarsky et al. (2019) by replacing the temporal component with a community variable to compare language use across Reddit communities. Lucy and Bamman (2021) also dive into differences between Reddit communities, but explore the possibilities of using contextualized models following an approach similar to the Word Sense Induction technique of Giulianelli et al. (2020). These approaches focus on clear denotation shifts.

More subtle shifts in connotation rather than denotation have hardly been explored. Basile et al. (2022) propose a system to automatically detect subtle shifts in connotation using static embeddings, indicating the potential to extract them from distributional information. Their focus is on shifts along a single connotation 'dimension' such as polarity. In contrast, we focus on shifts that are hard to capture by a singular dimension, encompassing variances in attitudes and worldviews within specific political communities.

The most widely used evaluation dataset came with the first SemEval shared task on LSCD in 2020 (Schlechtweg et al., 2020) and contains a relatively small set of clear diachronic shifts in denotation (e.g. *plain*, which gained the meaning of aircraft next to the meaning of surface) for four languages. We conduct experiments on the binary LSCD task for English, which involves determining whether the meanings of 37 target words have changed or not between the time spans of 1810-1860 and 1960-2010. The results on this challenge indicate that overall, methods based on static representations outperform those relying on contextualized models. Nevertheless, given the state of the field, where incidentally contextualized representation methods also demonstrated superior performance (in another task as mentioned earlier), it is unclear what approach would perform best on our use case of subtle semantic shifts across political communities. To

the best of our knowledge, our study is the first to investigate the detection of such shifts in political discourse using a spectrum of methods based on static and contextualized models.

## 3 Task & Data

In this study, we investigate the use of language models for the detection of subtle semantic shifts between different social communities. We introduce the following objective to address this issue:

> **Task:** Given (i) a target word whose connotative meaning is assumed to differ between two communities, and (ii) two subcorpora representing (one of) the two communities in which the target word occurs, determine whether these subcorpora originate from the same community or different communities.

We address subtle semantic shifts through two use cases focusing on political discourse in Dutch (use case 1) and English (use case 2). We leverage user-generated web-data from Reddit, a social media platform that enables the formation of communities (subreddits) based on a wide range of topics and interests. We use political subreddits and examine how politically loaded terms undergo connotational semantic shifts across different communities. For example, the word *climate* carries the connotation of climate change being a hoax in right-wing discourse, while it tends to imply a genuine threat in left-wing discourse.

**Use case 1** We collect comments from two Dutch subreddits on opposing ends of the Dutch political spectrum: `Poldersocialisme` (PS), a left-wing community, and `Forum_Democratie` (FD), a right-wing community. We have collected all the existing comments from these two subreddits up until April 3, 2022, using the Pushshift API (Baumgartner et al., 2020).

**Use case 2** We have collected all the comments from the English subreddits `hillaryclinton` (HC) and `The_Donald` (TD) using a similar approach. We specifically focus on the year 2016, the year of the presidential elections between Donald Trump and Hillary Clinton. The statistics of all collected Reddit data is presented in Table 1 and more details are described in Appendix A.

In each of the use cases, we took one of the datasets, FD and TD respectively, and split it randomly into two subcorpora. This resulted in three

| Subreddit | Poldersocialisme | Forum_Democratie | hillaryclinton | The_Donald |
|---|---|---|---|---|
| Period | 30 May 2018 to 3 April 2022 | 16 Dec. 2017 to 3 April 2022 | 1 Jan. 2016 to 31 Dec. 2016 | 1 Jan. 2016 to 31 Dec. 2016 |
| Comments | 58 537 | 149 674 | 1 137 508 | 10 452 040 |
| Tokens | 3 077 786 | 8 526 494 | 48 902 320 | 341 901 610 |
| Average length | 53 | 57 | 43 | 33 |

Table 1: Statistics of the collected Reddit data for the two use cases. Tokens are based on BERT subword tokenization and average comment length is given in tokens.

subcorpora for each language: two from the same community (FD1 and FD2 for the Dutch case, and TD1 and TD2 for the English case) and one subcorpus from the opposing community (PS for Dutch and HC for English). This division allowed us to compare the detection of semantic shifts within the same community and between different communities.

**Target words**   The target words for the experiments were carefully selected to test for hypothesized shift in connotation between the communities. Around a hundred random examples from each subreddit were taken to identify main topics that users discussed, and related keywords that go along with it. Comments mentioning these keywords were extracted (rule-based script) and about 50 comments (randomly sampled) per keyword were further analyzed by the first author with consultation from an expert in political communication. The aim of the analysis was to grasp how the members of the opposing communities conceptualized the keywords. Other relevant keywords that popped up frequently in the analysis were added to the list of keywords; words and topics that did not show contrastive conceptualizations after analysis were omitted. Table 2 presents our final selection of target words, along with their frequency of occurrence in the three subcorpora for each use case. We present a summary of the specific hypotheses for each target word in Appendix B and a more detailed description of target word selection and hypothesis generation for the Dutch use case can be found in Hoeken (2022).

Our analysis focuses on the connotational aspects of a target word as perceived by different political communities. These differences can be observed with respect to distinct types of connotations. For example, the Dutch word *socialist* diverges solely in terms of *evaluative connotation*, with a negative evaluation among the right-wing community (FD) and a positive evaluation among the left-wing community (PS). In contrast, for the

concept of climate (particularly in reference to climate change), disparities between the two Dutch communities extend beyond evaluative connotations. FD members associate the concept with hysteria while PS members view it as a problem. We refer to this latter type as *associative connotation*.

Table 2 illustrates our division of target words into evaluative and associative connotation types, visually denoted by a horizontal line. It is important to acknowledge that this categorization relies heavily on expertise in political discourse as well as a certain degree of subjectivity.

**Evaluation data and metrics**   We conduct experiments on our use-cases by testing each method on the above outlined set of 12 target words for each language. For each target word, two test scenarios are provided: a) two subcorpora originating from different communities b) two subcorpora originating from the same community (control). This results in a total of 24 test instances, which we provide with a label being either "same-community" or "different-community". For each of these instances a method outputs a distance value, the graded outcomes. We convert the graded outcomes to binary values by comparing the distance values of the two test scenarios for a target word with each other, and assign "different community" to the largest value, and "same community" to the smallest. Subsequently, we evaluate the number of correct binary predictions as well as the correlation between the methods' graded distance values and the gold labels (converted to 1 and 0). The latter is computed by calculating the Pearson Correlation Coefficient using the SciPy library. It should be considered that this distance comparison approach is limited as even very small differences between representations can lead to different classifications. Just like the SemEval test set, our test set is relatively small. We employ additional validation strategies to gain deeper insights into what signals various methods of shift detection can pick up (see Section 4).

| Dutch targets | PS | FD1 | FD2 | English targets | HC | TD1 | TD2 |
|---|---|---|---|---|---|---|---|
| klimaat *climate* | 418 | 1 177 | 1 182 | abortion | 2 941 | 7 829 | 7 911 |
| vaccinatie *vaccination* | 50 | 499 | 467 | muslims | 2 722 | 40 654 | 41 285 |
| immigratie *immigration* | 175 | 840 | 777 | islam | 910 | 54 666 | 54 404 |
| vluchteling *refugee* | 251 | 393 | 382 | immigrants | 2 230 | 18 682 | 18 498 |
| media *media* | 95 | 250 | 237 | citizenship | 521 | 5 054 | 5 092 |
| belasting *taxes* | 590 | 607 | 599 | guns | 4 157 | 19 496 | 19 307 |
| overheid *government* | 1 355 | 1 847 | 1 850 | taxes | 6 362 | 18 528 | 17 040 |
| links *left* | 4 640 | 4 093 | 4 053 | income | 3 986 | 7 523 | 7 689 |
| rechts *right* | 2 382 | 2 968 | 3 019 | police | 2 776 | 30 100 | 30 191 |
| socialist *socialist* | 1 397 | 175 | 163 | liberals | 3 246 | 38 718 | 39 035 |
| liberaal *liberal* | 1 214 | 628 | 618 | iran | 1 299 | 7 978 | 7 692 |
| kapitalisme *capitalism* | 1 801 | 232 | 258 | syria | 1 515 | 9 329 | 9 319 |
| **Corpus size** | 2.3M | 3.2M | 3.2M | **Corpus size** | 38.8M | 133.6M | 133.7M |

Table 2: Target words and their subcorpus frequencies (based on lemmatized data). Middle horizontal line indicates boundary between connotation type that the target word mainly differs in, above = associative and below = evaluative.

## 4 Methods & Experimental Set-up

In this section, we discuss the methods we employ to address the task of detecting subtle semantic shifts as defined in the previous section. We evaluate all methods on our subtle connotative shift task. In addition, we test the performance of all methods on the SemEval 2020 task to examine their behavior on denotational shifts.

### 4.1 Models and methods

**Static representations**  We consider two traditional LSCD methods based on **static** word embedding models. First, we use Positive Pointwise Mutual Information (**PPMI**), a transparent count-based model, where we align subcorpora via column intersection (Hamilton et al., 2016a; Dubossarsky et al., 2017, 2019). Second, we use Skipgram with negative-sampling (**SGNS**) obtained from Word2Vec (Mikolov et al., 2013), which performed best on SemEval-2020 (Schlechtweg et al., 2020). We apply the Orthogonal Procrustes technique to align two SGNS-models trained on different subcorpora (Hamilton et al., 2016b). Following common practices, we lemmatize all words and use Cosine Distance (CD) to calculate the difference between target word representations. More details on both methods can be found in Appendix C.

**Contextualized Representations**  We examine whether feeding a contextualized model different types of information impacts its ability to reflect subtle shifts when comparing internal representations. For obtaining representations from a contextualized model, either pre-trained or fine-tuned on a specific task (to be elaborated hereafter), we feed

the model each comment in which a target word occurs[3] in a community-specific subcorpus. For each context we extract the model's hidden layers, average over them and select the target word's sentence position, resulting in a contextualized embedding of the target word. The pairwise cosine distance is computed between two sequences of contextualized embeddings from two different subcorpora, by calculating the pairwise cosine distances and taking the average over them i.e. the Average Pairwise Distance (APD) (Giulianelli et al., 2020).[4] Below, we provide a description of all the different models we utilize in this procedure (see Appendix C for details).

**Vanilla Models**  First, we test the inherent capability of a pretrained contextualized language model in detecting semantic shifts by using **BERT** models. We use the base version of BERT (Devlin et al., 2019) for English and BERTje (de Vries et al., 2019) for Dutch. We evaluate the impact of fine-tuning BERT further on the data from the same domain as the test data (**D-BERT**). For the Dutch use-case, we used the union of test corpora. For the English use-case, we fine-tuned on a random sample of 5.8 million Reddit comments spanning from December 2005 to March 2023. We mainly follow Giulianelli et al. (2020) in finetuning a BERT model with the Masked Language Modelling (MLM) objective. Our final language model is **XLM-R** (Conneau et al., 2020), which yielded

---

[3]Due to the models' input limitations comments exceeding 512 tokens are excluded.

[4]All models are implemented through Hugging Face's transformers library and for APD calculation we use torchmetrics.

high performance in previous work (Rachinskiy and Arefyev, 2022).

**Fine-tuned Models**   A second set of experiments explores the impact of task-specific fine-tuning. Following Rachinskiy and Arefyev (2022), we fine-tune XLM-R on word sense disambiguation (**WSD XLM-R**). This system is based on a Bi-Encoder Model (BEM) (Blevins and Zettlemoyer, 2020) that consists of a context encoder for generating target word representations and a gloss encoder for generating representations of WordNet glosses. These encoders are jointly trained on SemCor (Miller et al., 1994), an English WSD-dataset, to classify a target word sense based on similarity of its representation with the glosses. As mentioned before, the intuition behind their approach is that aspects on sense are emphasized in the representations. As our task emphasizes connotation, we also investigate the impact of fine-tuning on sentiment analysis, which should foreground evaluative connotative meaning. To this end, we utilize an XLM-R model fine-tuned on a set of unified sentiment analysis Twitter datasets in eight languages (**SENT XLM-R**) (Barbieri et al., 2022).

**Masked target prediction**   We propose an alternative approach exploiting model behavior rather than internal representations. We leverage the masked token prediction task, which aligns with the natural pretraining process of BERT and D-BERT. We extract token substitutions in all given contexts of a target by masking it and predicting candidates. For each instance, we compile the top 10 most probable substitutions. Next, we determine the frequency of unique tokens predicted as substitutes throughout all contexts within a specific subcorpus. We compare substitutions for a target word in different subcorpora by analyzing the difference in relative frequencies of token predictions that are part of the intersection of the two substitution sets. We measure this difference by computing the Jensen-Shannon Divergence (JSD).

Our approach differs from the masked token prediction approach of Arefyev and Zhikov (2020) in two ways: 1) they do not mask the target word T, but replace T by the pattern 'T and [MASK]' and generate top k substitutions for the mask token, 2) they don't compare the prediction frequencies of the substitutes, but cluster the total collection of built Bag-of-substitutes vectors and compare the frequencies of clusters across subcorpora.

## 4.2   Evaluation

**Predicting real shifts**   We test whether the methods can reflect the subtle semantic shifts using the test corpora and metrics presented in Section 3. For each target word, we assess whether our methods can correctly reflect a higher difference between target word occurrences originating from different communities compared to target word occurrences originating from the same community.

We also test all methods on the English data for the SemEval 2020 shared task on LSCD (Schlechtweg et al., 2020), comparing the outputs against both the gradual and binary gold values (using correlation and accuracy[5]). The task differs from ours, as it requires a binary classification for each word, distinguishing between those that have changed and those that have not. To determine this binary label, we adopt the common practice of employing the mean distance values as decision threshold. In contrast, our subtle shifts task assesses whether a method can distinguish whether word occurrences are taken from the same (no shift) or from different (shift) subcorpora. Thus, performance values cannot be directly compared across the two tasks.

**Predicting simulated extreme shifts**   Our initial experiments showed that methods using contextualized models underperform. We conduct an additional evaluation procedure to 1) test if these methods could detect extreme semantic shifts, as a sanity check and 2) explore whether the underperformance of these methods could be caused by an overly strong pretraining signal overshadowing weaker signals from new contextual information.

We simulate a radical denotation shift in which words acquire entirely new senses and lose old senses (partially inspired by Schütze (1998) and Shoemark et al. (2019)). We duplicate a subcorpus and replace all mentions of a *donor word* with a *recipient word*. For English, we replace all occurrences of *guns* with the recipient word *taxes*. The original mentions of the recipient word were replaced by [UNK] tokens. The recipient word has now 'acquired' the contexts of the donor word and 'lost' its original contexts in the duplicated subcorpus, i.e. we simulate that *taxes* now means guns. For Dutch, we replace all instances of the donor word *klimaat* (climate) by the recipient word *liberaal* (liberal) in the FD1 data. If the represen-

---

[5]Accuracy is calculated using the scikit-learn library

tations of a contextualized model are sensitive to context, the distance between the recipient word in its original context (*taxes* in its original context) and the recipient word in the donor context (*taxes* in the context of *guns*) should be similar to the original distance between the donor and recipient word in their original contexts. We assess whether the D-BERT method can reflect the simulated shift.

# 5 Results

In this section we first present the results on our main task of subtle semantic shift detection within our use-cases (5.1). In Section 5.2 we discuss the results on simulated extreme shifts, after which the insights generated by our mask prediction method are showcased (5.3). Next, we evaluate our methods on the SemEval 2020 shared task (5.4). Finally, Section 5.5 provides an in-depth analysis of errors made by contextualized representation methods.

## 5.1 Main task

The results of our methods on the detection of subtle semantic shifts in our Dutch and English use cases are presented in Table 3. The approaches relying on distance comparison between static representations (PPMI and SGNS) demonstrate strong performance in both Dutch and English, showing high correlations and perfect binary predictions. The contextualized representation methods display notably weak performance scores, except for a few methods that demonstrate better binary classification performance (i.e. D-BERT and XLM-R for English). These findings emphasize the efficacy of static representations in capturing subtle semantic shifts in both Dutch and English use cases, whereas contextualized representations, even when provided with various types of specialized fine-tuning information, fail to achieve similar effectiveness.

The masked target prediction approach proved highly effective in detecting subtle semantic shifts. Specifically, it outperformed all previous methods for English, and for Dutch, it outperformed all methods except for the PPMI-method in terms of correlation performance. We dive deeper into the added transparency of this approach in Section 5.3.

## 5.2 Simulated extreme shifts

We test our models' sensitivity to context by assessing whether they can reflect simulated extreme semantic shift (Section 4.2). The results are summa-

|  | Dutch | | English | |
| --- | --- | --- | --- | --- |
|  | pears.r | # correct | pears.r | # correct |
| *Static representations* | | | | |
| PPMI | 0,878 | 24 | 0,702 | 24 |
| SGNS | 0,726 | 24 | 0,890 | 24 |
| *Contextualized representations* | | | | |
| BERT | 0,091 | 16 | 0,012 | 16 |
| D-BERT | 0,004 | 14 | 0,081 | 22 |
| XLM-R | 0,084 | 12 | 0,026 | 22 |
| WSD XLM-R | 0,057 | 16 | 0,006 | 18 |
| SENT XLM-R | 0,091 | 16 | 0,013 | 20 |
| *Masked target prediction* | | | | |
| BERT | 0,738 | 24 | 0,923 | 24 |
| D-BERT | 0,780 | 24 | 0,929 | 24 |

Table 3: Performance of the methods on detecting subtle semantic shifts in Dutch and English use-cases. '# correct' refers to the number of correct binary decisions out of 24 test instances.

rized in Table 4. As expected, in the original condition, the pairwise distances of *taxes*-representations with each other are smaller than the pairwise distances between *taxes* and *guns*. If model representations are sensitive to context, the pairwise distances between the original and manipulated *taxes*-representations should be comparable to the distances between the original occurrences of *taxes* and *guns* and larger than the distances between the original occurrences of *taxes*. However, we observe the opposite; the distance between the original and manipulated *taxes*-representations is comparable to the distance between the original occurrences of taxes and considerably lower than the distance between the original occurrences of *taxes* and *guns*. This observation reveals the method's inability to detect extreme semantic shift, while simultaneously implying that minimal sensitivity of the model's representations to contextual variations could account for this outcome. We observe the same pattern in experiments using the pretrained variant of BERT as well as conducted with Dutch data (see Appendix D). These results diverge from the findings by Ethayarajh (2019), who investigated the context specificity of word representations in various contextualized models. They suggest that, to varying degrees depending on the layer and model chosen, that on the whole contextualized models produce context-specific representations.

## 5.3 Deeper insights with mask prediction

We demonstrate the transparency of the masked target prediction method by analyzing the predicted

| Condition | Target Rep. | APD |
|---|---|---|
| original (recipient - donor) | taxes – guns | 0.715 |
| original (recipient - recipient) | taxes – taxes | 0.242 |
| simulated shift (recipient - recipient in donor context) | taxes – taxes | 0.251 |

Table 4: APD between representations in original and manipulated setting, where *guns* (donor) is replaced by *taxes* (recipient).

| Target | FD | PS |
|---|---|---|
| immigratie | "veiligheid", "werkloosheid" | "regime", "Westen" |
| kapitalisme | "marktwerking", "vrijheid" | "mensen", "vrouwen" |
|  | **TD** | **HC** |
| citizenship | "nationality", "status" | "equality", "immigration" |
| income | "money", "property" | "social", "gender" |

Table 5: Examples of predicted target word substitutions that demonstrate high disparity in relative frequency between two opposing communities

|  | SemEval-2020 | |
|---|---|---|
|  | **pears.r** | **acc.** |
| *Static representations* | | |
| PPMI | 0,137 | 0,568 |
| SGNS | **0,649** | 0,649 |
| *Contextualized representations* | | |
| BERT | 0,521 | **0,676** |
| D-BERT | 0,518 | 0,649 |
| XLM-R | 0,458 | 0,568 |
| WSD XLM-R | 0,247 | 0,541 |
| SENT XLM-R | 0,413 | 0,541 |
| *Masked target prediction* | | |
| BERT | 0,542 | 0,486 |
| D-BERT | 0,466 | 0,541 |
| Best SemEval (SGNS based) | - | 0,622 |

Table 6: Results on SemEval data

substitutions that demonstrated the highest disparity in relative frequency between two opposing communities. Several examples are presented in Table 5, which illustrate how this analysis can provide insights into the different (associative) connotations of a target word.

For instance, for the target word *kapitalisme* (capitalism) in contexts originating from the FD community, highly deviating associated words (compared to the PS community) are *marktwerking* (market forces) and *vrijheid* (freedom). Similarly, *werkloosheid* (unemployment) is a distinguishing association with the target word *immigratie* (immigration). In the English use-case *social* and *gender* emerge as words that BERT strongly associates with the target word *income* in the HC community, when compared to the TD community. The same goes for the association of *equality* with the target word *citizenship*. These observations align with our hypotheses and underscore the effectiveness of our approach relying on model behavior to detect connotative shifts.

### 5.4 Evaluation on SemEval

This section presents the results of our methods on the SemEval 2020 shared task on LSCD for English (Schlechtweg et al., 2020), as summarized in Table 6. As a basis for comparison, we provide the score of the best-performing system of the challenge (SGNS-based) in Table 6. The purpose of the SemEval evaluation is to compare the performance of our methods on a widely used evaluation set for clear diachronic shifts. As explained in Section 4,

performance on this task cannot be compared directly to our subtle shift task.

In terms of correlation, our SGNS method outperforms all other methods. This is consistent with the current state of LSCD research. In terms of accuracy, however, the BERT method (pretrained Vanilla) demonstrates the best performance, which differs from the findings of the SemEval results. This discrepancy could be explained by the fact that we used raw versions of the SemEval test data for the contextualized methods, whereas only lemmatized data was available for the shared task. This discrepancy highlights the importance of word form for contextualized models, as has also been emphasized in previous work (Laicher et al., 2021).

Moreover, we observe no noticeable improvement in performance on the SemEval test data when using the masked target prediction method, and fine-tuning XLM-R on downstream tasks did not yield improved results either. Regarding the WSD XLM-R method, which showed success in Spanish and Russian semantic shift tasks, our results imply that this method might not exhibit consistent performance across languages, or at least, across evaluation sets.

### 5.5 Error analysis

To delve deeper into the failures of contextualized representation approaches on our subtle shift dataset, we analyzed the specific error instances for each method (based on the binary pair classification, see Section 3). The misclassified words, listed in Table 11 in Appendix E, show that a high number of errors is shared by all methods. Overall,

methods seem to struggle with highly subtle shifts, which manifest in different ways.

The majority of the errors occur for words with evaluative shifts (e.g. *links* (left), *socialist* (socialist), *kapitalisme* (capitalism)), which are likely to be more subtle and thus more difficult to detect than associative shifts.

Some errors do, however, also occur for associative shifts. The misclassified words *overheid* (government), *belasting* (taxes) for Dutch and *taxes* for English could be argued to exhibit more nuanced shifts with some associations being shared by both communities, while others are fundamentally opposing. Both communities accept that the government provides support while sharing a level of criticism for it. Within the FD community, *overheid* is associated with a passive and limited role, whereas the PS community associates it with an active and larger role. In contrast, the associations of the concept of *klimaat* (climate) are much more distinct; it is associated with hysteria by the FD community and seen as a problem that needs to be solved by the PS community.

We analyze the occurrences of the word *islam*, which was misclassified by all methods. The original expectation posited an association of Islam with religion by the HC community compared to terrorism or illness in the TD community. We find that the corpus reflects a more nuanced picture of the shift; members of the HC community distinguish between Islam and radical Islam, with the latter receiving heavy criticism and sharing more similarities with TD standpoints. Within the HC community, there is extensive discussion about how the right perceives Islam, leading to similar language use as observed in the TD community. The shift can thus be characterized as a greater diversity of connotations within the HC community compared to a more consistent interpretation of *islam* within the TD community. This kind of distinction appears challenging for contextualized representation methods to pick up.

## 6   Discussion & Conclusion

We presented a systematic study of different methods using contextualized and static language models for detecting subtle semantic shift in political discourse in Dutch and English. Based on a limited but carefully selected test set, we found that methods based on static representations show strong performance in capturing these shifts, while con-

textualized representations, even when fine-tuned on specialized information such as domain or sentiment, fail.

Through simulation experiments with extreme semantic shifts, we revealed that representations extracted from pretrained contextualized models lack sensitivity to context, which contributes to their poor performance. We thus provide empirical evidence for speculations about the dominance of pretraining signals from previous work.

We explore the masked token prediction capabilities of contextualized models as an alternative and transparent method for subtle shift detection. Our approach relies on top k target word substitutions, and proved highly effective on our subtle semantic shift task. It allows for the analysis of predicted substitutions, which provided deeper insights into the different connotations attributed to the target word by different communities. Follow-up research could explore alternative values for k and use the prediction probabilities as additional signals. A threshold approach based on substitute probability could yield more informative substitutes, tuned for the targeted semantic shift type, and thus better performance on the SemEval data.

Our evaluation on the SemEval task revealed that model performance differs substantially in relation to the types of shifts under investigation. Our results on the SemEval task, which targets clear semantic changes in denotation over time, show that representations of pretrained BERT models can reflect such changes to a certain degree and underscore the importance of word form for contextualized models, as emphasized in prior research.

Our findings contribute to the understanding of semantic shifts in language and highlight the challenges and opportunities associated with the less-explored task of detecting subtle semantic shifts.

## Limitations

One of the prominent advantages that inherently come with BERT-like models is the ability to deal with out-of-vocabulary words using a subword tokenization. However, how to re-build the subword representations that preserve the compositional meaning of the original word is still an open-discussion. Representation-based approaches, which perform poorly on this task, can use simplistic approaches for reconstruction of the word such as subword-pooling (as in Bommasani et al., 2020), but the effectiveness of its results compared

to single-subword meaning representations remains unclear. The implications of this issue is potentially more problematic for morphology-rich languages. To mitigate this, our target words are selected from the union of the vocabularies of all methods and consist of one-subword-words.

We also acknowledge the importance of the number and selection of target words in evaluating the performance of semantic shift detection methods. Our results exhibited clear distinctions among the methods, but for future research with less significant differences, a larger and more diverse set of target words is recommended.

## Ethics Statement

An ethical concern of this type of research is that insights about perspectives arising from the corpus representative of a group are applied to individuals associated with the group. Care should be taken when interpreting tendencies based on language used by a Reddit community. The fact that a community talks about a concept in a certain way does not mean that an individual part of or sympathetic to the group shares this view. Furthermore, given the nature of forum posted data, it is important to protect authors' identities. For this research, only the content of posts was used and author information is not needed. We provide the code used for scraping the data. If we share data directly ourselves (upon request), it will only be shared in compliance with GDPR.

The domain that we have chosen to investigate subtle semantic shift, namely political discourse, is also prone to have stereotypical tendencies, that might have an effect on our empirical results. Moreover, it is also well-known to the NLP community that large pre-trained language models, which are trained on unstructured web-based text, inherently learn undesired stereotypical roles, therefore they might exhibit bias towards certain groups/ideas that can overlap with the content of our use-cases.

## Acknowledgements

We like to thank Mariken van der Velden, an expert in political communication, and Alina Rullkötter, a student assistent with a background in political science, for consultation on the conducted manual analyses of the Dutch and English use cases respectively.

The first two authors acknowledge financial support by the project "SAIL: SustAInable Life-cycle of Intelligent Socio-Technical Systems" (Grant ID NW21-059A), which is funded by the program "Netzwerke 2021" of the Ministry of Culture and Science of the State of Northrhine Westphalia, Germany. Antske Fokkens was supported by the EU Horizon 2020 project InTaVia: In/Tangible European Heritage - Visual Analysis, Curation and Communication (http://intavia.eu) under grant agreement No. 101004825.

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

## A  Data Collection

All subreddit comments posted within the specified time periods were collected using the Pushshift API, excluding: posts whose author is "AutoModerator" and post texts without any alphabetic character or deleted/removed posts (which only contain the text '[deleted]' or '[removed]'). For methods using contextualized models, we remove URLs and we replace some Reddit API inherent response body encodings with their characters (&, <, >). We lowercase text only if the pre-trained model used is trained on lowercased data (i.e. not BERTje). For methods using static models, we apply above preprocessing steps, lowercasing the text for all static models used, and additionally remove all non-alphanumeric characters and lemmatize the data using SpaCy.

## B  Hypotheses

Tables 7 and 8 summarize our hypotheses regarding the target words, which are derived from manual analysis of our datasets and expertise in political communication. A brief description is provided of the connotative aspect(s), either associative or evaluative (positive/negative), that the members of the community attribute to a target word, that differs from the opposing community.

| Targets | Forum_Democratie (FD) | Poldersocialisme (PS) |
|---|---|---|
| klimaat *climate* | hysteria | problem |
| vaccinatie *vaccination* | issue of freedom | protecting the vulnerable |
| immigratie *immigration* | massive, problem | responsibility, diversity |
| vluchteling *refugee* | problem, criminal | innocent, victim |
| media *media* | misinformation | information source |
| belasting *taxes* | theft | more on capital |
| overheid *government* | passive/limited role | active/bigger role |
| links *left* | negative | positive |
| rechts *right* | selective positive | negative |
| socialist *socialist* | negative | positive |
| liberaal *liberal* | selective positive | negative |
| kapitalisme *capitalism* | positive | negative exploitation |

Table 7: Dutch target words with hypothesized unshared connotative aspects

| Targets | The_Donald (TD) | hillaryclinton (HC) |
|---|---|---|
| abortion | women's right | murder |
| muslims | terrorists | victims |
| islam | terrorism, illness | religion |
| immigrants | criminals | pathway to citizenship |
| citizenship | privilege | right |
| guns | right to bear | problem |
| taxes | robbery | to raise on the wealthy |
| income | support current system | cause of inequalities |
| police | victim of violence | misconduct |
| liberals | negative | positive |
| iran | negative (policy) | positive (policy) |
| syria | negative (policy) | positive (policy) |

Table 8: English target words with hypothesized unshared connotative aspects

| Method | Details |
|---|---|
| PPMI | window size = 10, alpha = 0.75, k = 5 |
| SGNS | window size = 5, dimensions = 300, negative = 1, iters = 5, minimum frequency = 10 |

Table 9: Static representation methods - hyperparameters

## C  Method details

Details of the static representation methods and contextualized representation methods are given in Table 9 and Table 10, respectively.

| Method | Details |
|---|---|
| BERT (en) | *'bert-base-uncased'* |
| BERT (nl) | *'GroNLP/bert-base-dutch-cased'* |
| D-BERT (all) | epochs = 3, batch size = 8 |
| XLM-R | *'xlm-roberta-base'* |
| WSD XLM-R | epochs = 7, batch size = 8 |
| SENT XLM-R | *'cardiffnlp/xlm-roberta-base -sentiment-multilingual'* |

Table 10: Contextualized models: Hugging Face's models used and hyperparameters for fine-tuning. All other hyperparameters were set to the default values

## D  Manipulation Experiments

**English Use cases.**   The following list displays APD values between D-BERT target word representations in original and manipulated version of TheDonald subset 1.

- $\text{guns}_{\text{original}} - \text{guns}_{\text{original}}$ : 0.214
- $\text{taxes}_{\text{original}} - \text{taxes}_{\text{original}}$ : 0.242
- $\text{taxes}_{\text{original}} - \text{guns}_{\text{original}}$ : 0.715
- $\text{guns}_{\text{original}} - \text{taxes}_{\text{manipulated}}$ : 0.692
- $\text{taxes}_{\text{original}} - \text{taxes}_{\text{manipulated}}$ : 0.251
- $\text{taxes}_{\text{manipulated}} - \text{taxes}_{\text{manipulated}}$ : 0.228

The following list displays APD values between pretrained BERT target word representations in original and manipulated version of TheDonald subset 1.

- $\text{guns}_{\text{original}} - \text{guns}_{\text{original}}$ : 0.203
- $\text{taxes}_{\text{original}} - \text{taxes}_{\text{original}}$ : 0.224
- $\text{taxes}_{\text{original}} - \text{guns}_{\text{original}}$ : 0.718
- $\text{guns}_{\text{original}} - \text{taxes}_{\text{manipulated}}$ : 0.689
- $\text{taxes}_{\text{original}} - \text{taxes}_{\text{manipulated}}$ : 0.221
- $\text{taxes}_{\text{manipulated}} - \text{taxes}_{\text{manipulated}}$ : 0.188

**Dutch Use cases.**   The following list displays APD values between pretrained BERT target word representations in original and manipulated version of Forum_Democratie subset 1. Manipulation: *liberaal* in original contexts of *klimaat*.

- $\text{klimaat}_{\text{original}} - \text{klimaat}_{\text{original}}$ : 0.213
- $\text{liberaal}_{\text{original}} - \text{liberaal}_{\text{original}}$ : 0.222
- $\text{liberaal}_{\text{original}} - \text{klimaat}_{\text{original}}$ : 0.660
- $\text{klimaat}_{\text{original}} - \text{liberaal}_{\text{manipulated}}$ : 0.631
- $\text{liberaal}_{\text{original}} - \text{liberaal}_{\text{manipulated}}$ : 0.254
- $\text{liberaal}_{\text{manipulated}} - \text{liberaal}_{\text{manipulated}}$ : 0.237

The following list shows APD values between D-BERT target word representations in original and manipulated version of Forum_Democratie subset 1.

- $\text{klimaat}_{\text{original}} - \text{klimaat}_{\text{original}}$ : 0.175
- $\text{liberaal}_{\text{original}} - \text{liberaal}_{\text{original}}$ : 0.175
- $\text{liberaal}_{\text{original}} - \text{klimaat}_{\text{original}}$ : 0.644
- $\text{klimaat}_{\text{original}} - \text{liberaal}_{\text{manipulated}}$ : 0.643
- $\text{liberaal}_{\text{original}} - \text{liberaal}_{\text{manipulated}}$ : 0.213
- $\text{liberaal}_{\text{manipulated}} - \text{liberaal}_{\text{manipulated}}$ : 0.203

## E  Error Analysis

Table 11 lists all error instances for each method based on contextualized representations.

| | Dutch | English |
|---|---|---|
| BERT | immigratie, links, rechts, socialist | taxes, muslims, islam, syria |
| D-BERT | links, rechts, socialist, liberaal, kapitalisme | islam |
| XLM-R | belasting, overheid, links, rechts, socialist, kapitalisme | islam |
| WSD XLM-R | overheid, links, rechts, kapitalisme | islam, iran, syria |
| SENT XLM-R | belasting, rechts, socialist, kapitalisme | islam, syria |

Table 11: Error instances