# OpenReview forum: "Methodological Insights in Detecting Subtle Semantic Shifts with Contextualized and Static Language Models"
_EMNLP/2023/Conference — EMNLP 2023 Findings_

### Official Review · Reviewer_af38 · 2023-08-04

**Soundness:** 3

**Excitement:**

3: Ambivalent: It has merits (e.g., it reports state-of-the-art results, the idea is nice), but there are key weaknesses (e.g., it describes incremental work), and it can significantly benefit from another round of revision. However, I won't object to accepting it if my co-reviewers champion it.

**Paper Topic And Main Contributions:**

This paper explores how to automatically detect subtle semantic shifts between social communities. The focus is on political discourse from Reddit in English and Dutch. The authors perform an analysis of pre-existing methods that uses static and contextualized embeddings. In addition, the authors propose an approach using masked token prediction. They also do experiments with fine-tuning models on domain detection and sentiment analysis.

The authors show that methods using contextualized representation are not able to detect subtle semantic shifts. They have also tested the best performing systems in a Russian and Spanish shared task, and showed that this model did not perform well for detecting subtle shifts. Finally, they show that the masked token prediction approach is able to detect subtle semantic shifts.

The paper is clear and well written. The experiments are clearly discussed and described. The results are also well discussed. However, I am not an expert in the semantic shift field. The main weakness of the paper is the lack of discussion of the used dataset. There are very little details about the Reddit data, how it was created, processed, and what kind of ethical issues might rise from using and collecting this dataset.


**Reasons To Accept:**

- Subtle semantic change detection is a difficult task, and I enjoyed reading the various approaches tested by the authors.
- Good analysis and discussion of the results.
- Well written and clear paper.

**Reasons To Reject:**

- Too little information about the dataset used.
- No discussion about the ethical concerns related to gathering the data directly from Reddit (I'm thinking of GDPR and personal data).

**Reproducibility:**

3: Could reproduce the results with some difficulty. The settings of parameters are underspecified or subjectively determined; the training/evaluation data are not widely available.

**Reviewer Confidence:**

2: Willing to defend my evaluation, but it is fairly likely that I missed some details, didn't understand some central points, or can't be sure about the novelty of the work.

**Typos Grammar Style And Presentation Improvements:**

- Table 2: Performance of the methods on the detecting subtle semantic shifts in Dutch and English use-cases --> ... on detecting subtle ...

---

> ### Author Rebuttal · Authors · 2023-08-28
>
> Thank you for the review. We agree that the points raised need to be addressed and we are confident we can do so in a final version of the paper.
>
> **Point 1 (data)**
>
> We’ll add the most prominent information in the main text and details in the appendix. See rebuttal to reviewer j7jk, 2nd review, (dataset statistics) and reviewer ykvj , posted as review 1, (data selection process) for details on what will be added.
>
> **Point 2**
>
> Agreed, we should have been explicit about this. We will include it in the ethics section of the paper. Specifically, we plan to include the following text in the ethics section:
>
> *An ethical concern of this type of research is that insights about perspectives arising from the corpus representative of a group are applied to individuals associated with the group. Care should be taken when interpreting tendencies based on language used by a reddit community. The fact that a community talks about a concept in a certain way does not mean that an individual part of or sympathetic to the group shares this view.
> Furthermore, given the nature of forum posted data, it is important to protect authors’ identities. For this research, only the content of posts was used and author information is not needed. We provide the code used for scraping the data. If we share data directly ourselves (upon request), it will only be shared in compliance with GDPR.*

---

### Official Review · Reviewer_j7jk · 2023-08-04

**Soundness:** 4

**Excitement:**

3: Ambivalent: It has merits (e.g., it reports state-of-the-art results, the idea is nice), but there are key weaknesses (e.g., it describes incremental work), and it can significantly benefit from another round of revision. However, I won't object to accepting it if my co-reviewers champion it.

**Paper Topic And Main Contributions:**

The goal of this paper is to study the differences between static and contextualized word representations in a controlled fashion in their ability to detect subtle connotation shifts in language. This paper studies the problem by identifying two communities and a set of words that have different connotations in the two communities. It then uses word representations from various models to classify whether a given document containing one of these words belongs to one community or the other. The data itself is collected from subreddits belonging to two ends of the political spectrum, while models used range from static (skip-gram/Word2Vec, PPMI), contextualized (BERT, XLM-R, etc), as well as contextualized representations fine-tuned on a related tasks (word sense disambiguation and sentiment analysis). A separate approach leverages predicting a masked word (rather than classifying the document) using masked language models.

Experiments reveal that static representations are more accurate at detecting subtle semantic shifts and that contextualized models are unable to override the strong pre-training signals (also verified via a set of synthetic experiments). However, the masked target prediction approach is largely the most accurate one across the board. It is also observed that the fine-tuning on related tasks is not consistently helpful.

**Reasons To Accept:**

* Very thorough evaluation comparing various models in a controlled fashion. What was refreshing is that the paper systematically studies both static as well as contextualized embeddings, motivated by prior observations that static embeddings are still more accurate on this task. The analysis on the synthetic task to highlight the failures of the contextualized models was also illuminating (if a little surprising) and hence is a major contribution of this work.
* Overall, clearly written and well supported with meaningful analyses experiments.

**Reasons To Reject:**

* There is a distinct lack of information about the data used and collected. What is the nature of the data? What are some key statistics (size, number of documents, distribution)? Without some of this information, the dataset is a bit abstract and hence I am unable to draw conclusions about the results.

**Reproducibility:**

3: Could reproduce the results with some difficulty. The settings of parameters are underspecified or subjectively determined; the training/evaluation data are not widely available.

**Reviewer Confidence:**

3: Pretty sure, but there's a chance I missed something. Although I have a good feel for this area in general, I did not carefully check the paper's details, e.g., the math, experimental design, or novelty.

---

> ### Author Rebuttal · Authors · 2023-08-28
>
> Thank you for the review. We agree with the criticism, which we think can easily be fixed with the extra page and time for camera ready, in case of acceptance. We provide the details of the dataset statistics below. The extra space should leave room to add a table on the statistics as well as room to move at least one of the tables with target terms and predictions from the appendix to the main text, which should make the story less abstract. Other details will be provided in the appendix.
>
> ### Details:
>
> **General information on subreddits:**
>
> * Poldersocialisme: members supporting the Dutch left political movement that includes progressive parties such as the SP (Socialist Party) and GroenLinks (`GreenLeft’’).
> * Forum_Democratie: members supporting the Dutch right party Forum voor Democratie (Forum for Democracy).
> hillaryclinton: members supporting Hillary Clinton (Republicans).
> * The_Donald: members supporting Donald Trump (Democrats), banned since June 2020.
>
> **Dataset statistics**
>
> | Subreddit                   | Poldersocialisme            | Forum_Democratie            | hillaryclinton              | The_Donald                  |
> |-----------------------------|-----------------------------|-----------------------------|-----------------------------|-----------------------------|
> | Period                      | 30 May 2018 to 3 April 2022 | 16 Dec. 2017 to3 April 2022 | 1 Jan. 2016 to 31 Dec. 2016 | 1 Jan. 2016 to 31 Dec. 2016 |
> | Comments                    | 58 537                      | 149 674                     | 1 137 508                   | 10 452 040                  |
> | Tokens*                     | 3 077 786                   | 8 526 494                   | 48 902 320                  | 341 901 610                 |
> | Average length(in subwords) | 53                          | 57                          | 43                          | 33                          |
>
>
> *Tokens are based on BERT tokenization
>
> **Data scraping**
>
> * All subreddit comments posted within the specified time period were collected using the Pushshift API, excluding:
> posts whose author is “AutoModerator”
> * post texts without any alphabetic character or deleted/removed posts (which only contain the text ‘[deleted]’ or ‘[removed]’)
>
> **Data preprocessing**
>
> * For methods using contextualized models, we remove URLs and we replace some Reddit API inherent response body encodings (‘&amp;’, ‘&lt;’, ‘&gt;’) with their characters (&, <, >). We lowercase text only if the pre-trained model used is trained on lowercased data (i.e. not BERTje).
> * For methods using static models, we apply above preprocessing steps, lowercasing the text for all static models used, and additionally remove all non-alphanumeric characters and lemmatize the data using SpaCy.
>
> For more information about the data selection process, see our reply to ykvj (posted as 1st review).

---

### Official Review · Reviewer_ykvj · 2023-08-05

**Soundness:** 4

**Excitement:**

4: Strong: This paper deepens the understanding of some phenomenon or lowers the barriers to an existing research direction.

**Missing References:**

In addition to the reference cited above, you may be interested in the following:
* Cassotti et al. (2023) for another recent BERT-based method which obtains good results
* Miletic et al. (2021) as a further example of modeling subtle semantic shifts
* Gulordava and Baroni (2011), for a discussion of the fact that differences in context do not necessarily equate to differences in meaning, which may be relevant to your view of subtle changes in meaning (see p. 70 in their paper; you already cite it but with a broad focus)


Cassotti, P., Siciliani, L., DeGemmis, M., Semeraro, G., & Basile, P. (2023). XL-LEXEME: WiC pretrained model for cross-lingual LEXical sEMantic changE. Proceedings of ACL. https://aclanthology.org/2023.acl-short.135

Ethayarajh, K. (2019). How contextual are contextualized word representations? Comparing the geometry of BERT, ELMo, and GPT-2 embeddings. Proceedings of the EMNLP-IJCNLP. https://aclanthology.org/D19-1006/

Gulordava, K., & Baroni, M. (2011). A distributional similarity approach to the detection of semantic change in the Google Books Ngram corpus. Proceedings of GEMS. https://www.aclweb.org/anthology/W11-2508

Miletic, F., Przewozny-Desriaux, A., & Tanguy, L. (2021). Detecting contact-induced semantic shifts: What can embedding-based methods do in practice? Proceedings of EMNLP. https://aclanthology.org/2021.emnlp-main.847

**Paper Topic And Main Contributions:**

The paper evaluates static and contextualized word embedding methods on  lexical semantic change detection (LSCD), distinctly focusing on differences in connotative meaning across speaker communities as a subtle type of semantic shift. The authors use English and Dutch subreddits of opposing political views, and evaluate their methods on a manually curated list of 12 target words per language, containing items for which they expect connotative differences. They furthermore evaluate on diachronic English data from SemEval 2020, and as a general control simulate an extreme change in denotative meaning for one target item per language. Static word embeddings perform strongest overall; out of the contextualized approaches, a masking-based method introduced here is the strongest, but does not generalize well to SemEval data.

**Questions For The Authors:**

A. Taking into account comparisons between different pairs of corpora (same vs. different community), could you provide more details on (i) the precise experimental procedure; (ii) the way in which evaluation scores were computed?

B. Can you provide further details and examples regarding the selection of target items, especially when it comes to ensuring that the contexts in which they appear (i) are distributed as expected across the community-specific subreddits; (ii) are distinctive enough for the model to be reasonably expected to capture them?

C. Regarding the extreme semantic shift simulation, have you considered replacing target words with [MASK] token and taking its embeddings in different contexts, rather than replacing one target word with another? This would presumably avoid the contribution specific to the target word and more clearly reflect the effect of context.

**Reasons To Accept:**

* The paper provides a novel point of view on LSCD, focusing on a fine-grained and theoretically motivated type of semantic shift, much closer in nature to issues targeted by descriptive linguistic work. This focus brings much needed depth of analysis to the field, including through a detailed qualitative and error analysis.
* The proposed masking-based method is intuitive and performs well on the main dataset used in the study, warranting further experiments to optimize it on other datasets.
* The experimental setup is overall well thought through, with adequate comparisons across models and datasets.
* The analysis is conducted on two languages and hence not restricted to English.
* The paper is overall well written and easy to follow.

**Reasons To Reject:**

* The paper lacks important explanations on the precise experimental setup (extraction of embeddings and distance measurements across pairs of corpora) and especially the computation of evaluation metrics. This makes it difficult to reliably assess the reported results and limits reproducibility.
* The size of the evaluation dataset (12 target words per language) is on the extreme low end of LSCD datasets. While this is adequate for a much appreciated qualitative analysis, the authors use it for a quantitative assessment leading to broad performance claims. However, the size of the dataset calls into question the generalizability of the results, which is also suggested by comparisons with SemEval data.
* The extreme semantic shift simulation concludes that the contextualized embeddings of a target word in very different contexts remain more similar to one another than to those of a different word used in the same context. But this is not surprising, e.g. Ethayarajh (2019) reported the same general trend for BERT representations. Given established knowledge in the literature, I believe that these manipulations amount to a sanity check (as the authors also suggest) and should be presented less prominently.

**Update after the rebuttal:** Thank you for your very thorough response, which has fully addressed my concerns. Regarding the work by Ethayarajh (2019), I was referencing findings from Section 4.1 (paragraph on BERT, p. 60). Having re-read the paper, I may have overstated their relevance, but I still think your work would benefit from a comparison with those findings and a clearer framing of the semantic shift simulation experiment. I am happy to increase the soundness score.

**Reproducibility:**

4: Could mostly reproduce the results, but there may be some variation because of sample variance or minor variations in their interpretation of the protocol or method.

**Reviewer Confidence:**

4: Quite sure. I tried to check the important points carefully. It's unlikely, though conceivable, that I missed something that should affect my ratings.

**Typos Grammar Style And Presentation Improvements:**

Presentation suggestions:
* It would be helpful to provide examples and definitions of connotative change from the very beginning of the paper (top of the introduction).
* As indicated by the questions, I would suggest that you include further details on the experimental setup and recalibrate the importance of the section on semantic shift simulation.
* You may wish to acknowledge the size of your dataset more directly (e.g. l. 327).
* Table 1: consider normalizing the frequencies so they are comparable across the corpora.

Typos etc.:
* l. 11: that => which (or omit the comma)
* l. 30: social different => socially different
* fn 2, p. 2: Tahmasebia => Tahmasebi; also applies to the reference itself
* l. 385: approachis => approach is
* l. 446: extra "values"
* l. 505: full stop missing

---

> ### Author Rebuttal · Authors · 2023-08-28
>
> Thank you for the review. To summarize our rebuttal: we agree with the issues raised and think that we can fix most issues for camera ready (in case of acceptance). We plan to use the extra page for the most vital information on data and experimental setup, provide full details in appendices and adjust the way we present results on the use case data. For readability, we provide short answers to the questions and reactions to critical points first and more details further below.
>
> ### Answers to questions
>
> **Question A (Details of the experimental setup):**
>
> The details can be found at the bottom of the rebuttal. We agree that they should be in the paper and will add main information in the running text and details in appendices. We will also ensure that our published code and documentation are in line with the information provided in the paper.
>
> **Question B (Dataset selection):**
>
> Yes, we can provide more information and we checked the distribution of the examples over the community corpora (we provide details below).
>
>
> **Question C (semantic shift with [mask] token):**
>
> The suggested experiment could serve as an extra validation of our observation; based on the outcome of our synthetic shift experiments, we would expect the mask token approach to fail terribly. We conducted this experiment on the Dutch use-case with the pre-trained Dutch BERT model. The results confirm the above.
>
> ### Reactions to criticism
>
> **For point 1.** See our answers to question A and B and details below.
>
> **Point 2.**
>
> The use case specific evaluation sets are indeed on the small side and we agree that an overall quantitative assessment risks being misinterpreted. The purpose of the 24 terms (12 per language) is to explore the validity of results given a realistic use-case scenario (as opposed to a large-scale evaluation dataset). As such, the  terms represent core concepts of a specific political discourse and their selection is informed by political expertise. We observe that standard methods using contextualized models cannot represent fundamental differences in perspectives for any of the 12 targeted concepts (visible in the extremely low Pearson correlations) neither in Dutch nor English. We realize that the way we represent the results is not the best way of showing a clear outcome based on a small dataset. We plan to rectify this by showing results broken down to the individual terms.
> We provide more details on the process of finding the target terms below.
>
> **Point 3.**
>
> We agree that showing the relation between Ethayarajh (2019) and our work would strengthen the paper. As far as we understand Ethayarajh’s paper, it shows that word representations extracted from contextualized models do carry context-specific information (in particular the experiment presented in Section 4.3 - Static v.s. contextualized).  We would greatly appreciate further clarification if we are misunderstanding Ethayarajh (2019).
>
> Our motivation for giving prominence to the outcome of the extreme semantic shift simulation is to foster awareness in the semantic change community. We would therefore like to keep it prominent, but will of course make sure that it is clear that this is in line with earlier findings, if the same tendency has indeed been found in earlier work.
>
> Thank you for pointers to additional references and to p. 70 in Gulordava and Baroni (2011). They will improve the paper.
>
> ### Details
>
> **Question A (details of the experimental setup and evaluation metrics):**
>
> **Embedding extraction**
>
> Static embeddings: We create PPMI and SGNS embeddings using the hyperparameters stated in the appendix for each of the corpora.
>
> For the extraction of contextualized embeddings and distance measurements we mainly follow the procedure outlined in Kutuzov & Giulianelli (2020).
>
> For each target word, we feed the model, either the pre-trained model (BERT, BERTje or XLM-R) or the model fine-tuned on a specific task (to be elaborated hereafter), each comment in which the target word occurs (due to the models’ input limitations comments exceeding 512 tokens are excluded) in a community-specific subcorpus. For each context we extract the model’s hidden layers, average over them and select the target word’s sentence position, resulting in a contextualized embedding of the target word.
>
> **Distance measures**
>
> The pairwise cosine distance is computed between two sequences of contextualized embeddings from two different subcorpora, by calculating the pairwise cosine distances and taking the average over them i.e. the average pairwise distance (APD). All models are implemented through Hugging Face’s transformers library and for APD calculation we use torchmetrics.
>
> **Evaluation scores**
>
> We conduct experiments on our use-cases by testing each method on a set of 12 target words for each language. For each target word, two test scenarios are provided: a) two subcorpora originating from different communities b) two subcorpora originating from the same community (control). This results in a total of 24 instances, which we provide with a label being either "same-community'' or "different-community''. For each of these instances a method outputs a distance value, the graded outcomes. We convert the graded outcomes to binary values by comparing the distance values of the two test scenarios for a target word with each other, and assign “different community” to the largest value, and “same community” to the smallest. So, the outcomes for e.g. the PPMI-method on the English use-case look like:
>
> |    | target      | gold label          | method distance | method binary prediction |
> |----|-------------|---------------------|-----------------|--------------------------|
> | 1  | abortion    | different-community | 0,838           | different-community      |
> | 2  | abortion    | same-community      | 0,702           | same-community           |
> | 3  | citizenship | different-community | 0,816           | different-community      |
> | 4  | citizenship | same-community      | 0,725           | same-community           |
> | …  | …           | …                   | …               | …                        |
> | 23 | taxes       | different-community | 0,853           | different-community      |
> | 24 | taxes       | same-community      | 0,761           | same-community           |
>
>
> Subsequently, the correlation between the methods’ (graded) distance values and the gold labels (converted to 1 and 0) are computed by calculating the Pearson Correlation Coefficient using the SciPy library. For the classification performance, we calculate the accuracy score comparing the gold labels with the method’s binary predictions, using the scikit-learn library.
> As mentioned above, we plan to adjust the presentation of the results to address your comment.
>
> **Fine-tuning the models:**
>
> D-BERT: we again mainly follow Kutuzov & Giulianelli (2020) in finetuning a BERT model with the Masked Language Modelling (MLM) objective on data from the same domain as the test data. For the Dutch use-case, we used the union of test corpora. For the English use-case, we fine-tuned on a random sample of 5.8 million Reddit comments spanning from December 2005 to March 2023. During fine-tuning, input instances were padded, i.e. supplemented with [PAD] tokens, to ensure identical instance lengths of 512 tokens. A model was fine-tuned for 3 epochs with a batch size of 8 instances. All other training hyperparameters were set to the default values, as also implemented by Kutuzov & Giulianelli (2020).
>
>
> WSD-XLMR: We exactly follow the methodology presented by Rachinskiy and Arefyev (2022). This system is based on a Bi-Encoder Model (BEM) (Blevins & Zettlemoyer 2020} that consists of a context encoder for generating target word representations and a gloss encoder for generating representations of WordNet glosses. These encoders are jointly trained on SemCor (Miller et al. 1994), an English WSD-dataset, to classify a target word sense based on similarity of its representation with the glosses. We finetune for 7 epochs with a batch size of 8. All other training hyperparameters were set to the values as also implemented by Blevins & Zettlemoyer (2020).
>
> **Question B (data selection details):**
>
> **Procedure of target word selection:**
>
> Around a hundred random examples from each subreddit were taken to identify main topics that users discussed, and related keywords that go along with it.
> Comments mentioning these keywords were extracted (rule-based script) and about 50 comments (randomly sampled) per keyword were further analyzed.
> The aim of the analysis was to grasp how the members of the opposing communities conceptualized the keywords. We can for one or more keywords add several comments from each subreddit to illustrate their contrastive conceptualization.
> Other relevant keywords that popped up frequently in the analysis were added to the list of keywords and the same approach was taken; words and topics that did not show contrastive conceptualizations after analysis were omitted.
> (Only for the Dutch use-case) To validate the findings and gain additional insights, consultation was sought from an expert in political communication.
>
> To answer the specific questions:
>
> (i) by taking a consistent approach that includes random sampling, we aim to ensure to capture a distribution of contexts that is representative for the subreddit, although we acknowledge that the limited sample sizes might not fully ensure this.
>
> (ii) Distinctiveness of contexts: It is difficult to measure distinctiveness in absolute terms. Beyond qualitative checks, we can see that the PPMI models can reflect the difference between the two corpora. As PPMI directly reflects contexts, a clear distinction by PPMI indicates that the contexts are distinctive. Full control over distribution and distinctiveness of contexts in the two corpora would require artificial data selection, creation or manipulation. Our substitution experiments are an example of this.
>
> We provide more details on data statistics in our answer to reviewer j7jk (posted as the 2nd review).

---

### Meta-Review · Area_Chair_G6Qx · 2023-09-10

**Recommendation:** 5

**Metareview:**

The reviewers agree the paper provides a thorough evaluation of static and contextualized embeddings in lexical semantic change detection.
They greatly approve the depth of analysis and the experimental setup, especially on the synthetic task.

For the camera-ready version, the authors should:
1) Include detailed information about their datasets
2) Add discussion about ethical concerns of using Reddit data
3) Include explanation of the relation to experiments from Ethayarajh (2019).

---

### Decision · Program_Chairs · 2023-10-07

**Decision:**

Accept-Findings

**Comment:**

The reviewers agree the paper provides a thorough evaluation of static and contextualized embeddings in lexical semantic change detection.
They greatly approve the depth of analysis and the experimental setup, especially on the synthetic task.

For the camera-ready version, the authors should:
1) Include detailed information about their datasets
2) Add discussion about ethical concerns of using Reddit data
3) Include explanation of the relation to experiments from Ethayarajh (2019).